# Acceptability of unsupervised peer-based distribution of HIV oral self-testing for the hard-to-reach in rural KwaZulu Natal, South Africa: Results from a demonstration study

**Marcel K. Kitenge**[1,2]*, **Chinmay Laxmeshwar**[1], **Elkin Bermudez Aza**[3], **Ellie Ford-Kamara**[1], **Gilles Van Cutsem**[4,5], **Ntombi Gcwensa**[1], **Esther C. Casas**[4], **Khanyo Hlophe**[6], **Petros Isaakidis**[4], **Liesbet Ohler**[1]

**1** Médecins Sans Frontières (MSF), Eshowe, KwaZulu Natal, South Africa, **2** Division of Epidemiology and Biostatistics, Department of Global Health, Faculty of Medicine and Health Sciences, Stellenbosch University, Cape Town, South Africa, **3** Médecins Sans Frontières (MSF), International Office, Amsterdam, Netherlands, **4** Southern Africa Medical Unit, Médecins Sans Frontières (MSF), Cape Town, South Africa, **5** Centre for Infectious Disease Epidemiology and Research, University of Cape Town, Cape Town, South Africa, **6** Department of Health, King Cetshwayo District, KwaZulu Natal, South Africa

* msfocb-eshowe-epi@brussels.msf.org, marcel.kanyinda@gmail.com

## Abstract

### Background

Innovative models to distribute oral HIV self-tests (HIVST) provide an opportunity to increase access to HIV testing, especially for hard-to-reach populations. This study aimed to describe the acceptability of unsupervised peer-distribution of HIVST as a method to scale-up HIV testing.

### Methods

In this study, lay counsellors or community health workers provided HIVST kits to primary recipients (PRs) for distribution to their sexual partners, anyone in their social network (termed secondary recipients) or for self-testing, from September 2018 to March 2020. The study was conducted in Eshowe and Mbongolwane areas in KwaZulu-Natal, South Africa. A structured questionnaire was administered during the recruitment and passive follow-up, when people came for confirmatory HIV testing. Electronic records were retrospectively examined to determine initiation of antiretroviral treatment (ART) for all HIVST users and non-users.

### Results

Among 36,708 people approached to be primary recipients, 9,891 (26.9%) accepted; 31,341 HIVST kits were distributed with a median of three (IQR: 2–4) per peer. PRs were predominately recruited at primary health clinics (PHCs). However, acceptability of HIVST was thrice as high at community-based testing sites compared to PHCs (64.5% vs. 21.0%; p<0.001). During the study period, 34,715 adults were tested for HIV at both PHCs and

---

**Data Availability Statement:** First and second de-identified recipients databases, including all

variables are available on request in accordance with MSF's data sharing policy. Requests for access to data should be made to data. sharing@msf.org. For more information please see: A) MSF's Data Sharing Policy: https://www. msf.org/sites/msf.org/files/msf_data_sharing_ policycontact_infoannexes_final.pdf B) MSF's Data Sharing Policy PLOS Medicine article: http:// journals.plos.org/plosmedicine/article?id=10.1371/ journal.pmed.1001562.

**Funding:** Médecins sans Frontières provided funding for the study. All the authors except for KH are or were employed part-time or full-time by Médecins sans Frontières and received support in the form of salaries. received individual funding for participating in the running of the study or writing the article. All the authors did the work as part of their job. The funders had no role in study design, data collection and analysis, decision to publish, or preparation of the manuscript.

**Competing interests:** The authors have declared that no competing interests exist.

community-based testing sites; of these, 1,089 individuals reported HIVST use. Among HIVST users, 893 (82.0%) returned to the clinic for confirmatory testing after testing negative on HIVST; 196 (17.9%) were confirmed HIV positive following a positive HIVST. After excluding 36/196 (18.4%) participants for whom clinical records could not be found in electronic register and 25/160 (15.6%) who were already on ART before receiving HIVST, 129/ 135 (95.5%) initiated ART, whereas 2,362/2685 (88%) of HIV positive HIVST non-users-initiated ART.

## Conclusion

Unsupervised peer-distribution of HIVST was feasible and acceptable, with more than 25% accepting to be peer-distributors. Acceptability of HIVST was thrice as high in community sites compared to clinics.

## 1. Background

In 2019, more than two-thirds of people living with HIV (PLHIV) were in Africa [1]. World-wide, 19% of PLHIV did not know their status in 2019 [2], while in South Africa this figure was 8% [1]. The Joint United Nations Programme on HIV/AIDS (UNAIDS) has set the ambitious 95-95-95 goals with the aim to end the HIV/AIDS epidemic by 2030-stating that 95% of PLHIV should know their HIV status, among whom 95% should be on ART, and of those 95% should achieve viral suppression [3].

HIV self-testing (HIVST) has been recommended by the World Health Organization (WHO) as an additional testing method along with conventional healthcare worker-driven HIV testing [4]. Growing evidence suggests that HIVST can overcome barriers (long waiting times, lack of confidentiality) associated with conventional HIV testing approaches, leading to increased uptake [5, 6]. Provision of HIVST may also help reach those who have never tested or are unwilling to visit a health facility [6].

By July 2018, 59 countries had adopted HIVST policies [7]. Following the development of HIVST guidelines by the Southern African HIV Clinicians Society [8], South Africa published a national guideline on self-testing in May 2018 [9].

HIVST kits can be distributed through various channels. Community-based, facility-based and partner delivery have been the most widely researched HIVST distribution strategies [10–13]. Evidence from South Africa has demonstrated that HIVST can be a feasible option to improve uptake of HIV testing services (HTS) at clinics, and the use of HIVST in an unsupervised environment can be feasible and the results so obtained can be reliable However, linkage to care and confirmatory testing and ART initiation for unsupervised HIVST was relatively low [14–16].

Existing literature suggests that promoting and offering free HIVST using several different models might lead to increase testing coverage, especially among people who do not have easy access to testing facilities to visit (hard-to-reach groups) [17]. Peer-based-distribution among general population might help reach more people than only targeted distribution. Young people and men usually prefer not to go to health facilities for HIV testing and other health services [18, 19]. Encouragement by peers can help them to overcome their reluctance to using health facility-based HIV services [20]. However, there is lack of research on unsupervised peer-distribution of HIVST in general population, especially under programmatic conditions.

Hence, we conducted this study to assess the acceptability of unsupervised peer-distribution of HIVST and ART initiation for participants who tested HIV-positive using HIVST.

## 2. Methods

### Design

This was a cross-sectional analytical study of peer-led, unsupervised distribution model of HIVST kits. Lay counsellors or community health workers provided HIVST kits to primary recipients (PRs) for distribution to their sexual partner, anyone in their social network (termed secondary recipients) or for self-testing. In addition, data was collected prospectively among those returning for confirmatory testing.

### Study setting

Eshowe and Mbongolwane area in KwaZulu-Natal, South Africa, are mostly rural with one semi-urban market town. The population of the area was 114,490 in 2011 [21], when Médecins Sans Frontières (MSF), in partnership with the KwaZulu-Natal Department of Health, started the "Bending the Curves" project. A population-based survey conducted by MSF reported that the project catchment area had exceeded the UNAIDS 90-90-90 targets by achieving 90-94-95 in 2018. The survey also reported that the HIV prevalence was 18.5% in men and 30.5% in women and peaked at 50.5% among women aged 35–39 years [22].

### Study procedures

From September 2018 to February 2020, OraQuick Self-Test (OraSure Technologies Inc), an oral-fluid based test kit, was offered at 10 MSF-supported Department of Health primary health clinics and at 10 MSF community-based testing sites (called Luyanda sites) in rural areas. Luyanda sites serve communities with limited access to health facilities. At these sites, trained community health workers (CHWs) conduct HIV testing, TB screening and sample collection, condom distribution, random blood glucose testing for screening of diabetes mellitus and blood pressure monitoring for hypertension screening. CHWs also conduct TB and HIV education.

### Operational definitions

- Hard-to-reach groups include sub-groups of the population that may be difficult to reach through conventional HIV testing services and include those who have no easy access to testing facilities and men, adolescents and young adults.

- Primary recipients (PRs) are individuals who received HIVST kits from health care providers at primary health clinics (PHCs) or community-based testing sites for peer-distribution and/or self-testing.

- Secondary recipients (SRs) are those who received the test from a PR and use the test. Individuals identified at the HIV testing facilities who reported use of HIVST could be either PR or SR.

- Acceptability was defined as the proportion of people invited to be PRs and agreed to take HIVST kits for use and distribute them to their sexual partner, family members or anyone in their social network.

## Recruitment

PRs were recruited at the study sites using strategies designed to suit the context of the two types of study sites. At PHCs, lay counsellors introduced the study in the waiting areas and invited all users to participate. At community-based testing sites, CHWs introduced the study to all individuals visiting these sites, irrespective of the reason for their visit. Individuals who expressed interest were given detailed information about the study and screened for eligibility. After informed consent, they were enrolled in the study as PRs and given between 2–5 HIVST kits, based on their choice, for distribution and/or self-testing. PRs were at least 18 years of age and recruited irrespective of their self-reported HIV status. Given the emotional and psychological distress associated with a new HIV-positive diagnosis, those diagnosed with HIV on the day of visit were excluded.

Each kit was packaged in a paper bag with an information leaflet describing how to use HIVST and a care card with a unique study number and unique test number. HIV counsellors and CHWs, at PHCs and community-based testing sites, respectively, provided a brief explanation about how to use the test kits and what to explain to SRs before handing out HIVST kits. PRs were instructed to distribute HIVST kits to their sexual partner, family members or anyone in their social network who was above 18 years old. They could also test themselves if they were HIV-negative. PRs were instructed clearly that the test should not be used by PLHIV on ART. They were instructed to explain to the SRs to visit a clinic or a community-based testing site for confirmatory testing in case they had a HIVST positive. The care card provided with each test kit had a toll-free contact number where the HIVST users could call in case they wanted to talk to a counsellor.

A key component of testing in the study was ensuring confidentiality. SRs who ultimately received the HIVST kits were neither mandated to self-test in the presence of the PRs nor required to disclose their results to the PRs. After handing over the tests to the PRs, we did not actively follow them up. HIVST users would decide on their own agency to visit the clinic for confirmatory testing and ART initiation.

## Passive follow-up

During the study period, everyone coming to the study sites for HIV testing was asked if they had used HIVST and from whom they received an HIVST. In case they reported using HIVST, with or without presenting the care card, they were informed about the study and asked for consent to be enrolled in the study. Following the consent process, the study counsellor conducted individual face-to-face data collection using a structured questionnaire. A confirmed HIV-positive was defined as two consecutive positive HIV rapid tests by Determine™ (Alere, Waltham, USA) and Uni-Gold™ (HIV Trinity Biotech, Bray Ireland).

## Data collection and management

At the study sites, clinic staff and CHWs used a register to record the number of people invited to participate in the study and the number of those who accepted to be PRs. Data on the PRs were collected through a structured questionnaire. For the individuals who used HIVST and who returned at the health facility, after obtaining consent, the study site staff would fill in a structured proforma. All data were entered in an Epi Info 7 (CDC, Atlanta, USA) database.

Linkage to care data for HIVST users and non-users who were confirmed HIV-positive through HIV testing services (HTS) from the study sites was extracted from the South African national HIV electronic register (tier.net). ART initiation was checked three months after a positive test.

## Data analysis

Continuous variables were summarised using mean and standard deviation if normally distributed, otherwise median and interquartile range were used categorical variables were summarised using percentages or proportions. Pearson's Chi-squared test and Fisher's exact test were used to assess the statistical significance of differences in categorical outcomes between selected groups. Two-tailed tests were used throughout and the level of significance was set at 5%. All analyses were performed using STATA 16 (StataCorp LLC, 2019, College Station).

## Ethical considerations

Ethics approval was obtained from the University of Cape Town Human Research Ethics Committee, Cape Town, South Africa (HREC 080/2018) and the MSF Ethics Review Board, Geneva Switzerland (Protocol 1817). Permission to conduct the study in the public health facilities was granted by Department of Health, King Cetshwayo District. All PRs and SRs provided written informed consents before undergoing the study procedures and information such as age, gender, number of tests were collected. We did not collect any identifying information such as name, surname, address, phone number,or ID numbers.

## 3. Results

### Distribution of HIVST: Recruitment of PRs, acceptability and participant characteristics

Between September 2018 and March 2020, 36,708 individuals were approached to participate in the study as PRs, of among whom 9,891(26.9%) accepted. *Among the 9,891 who accepted*, 2,973(30.0%) were men and 6,918(70.0%) were female. At community-based testing sites 3,293/5181 (63.5%) accepted to be PRs, while at PHCs 6,598/31,527 (21.0%) accepted to be PRs. A total of 31,341 HIVST kits were distributed, among which, 10,643 (34.0%) were distributed at community-based testing sites and 20,698 (66.0%) distributed at PHCs (**Table 1**) with a median of three [interquartile range (IQR: 3–4)] kits distributed per PR. Acceptability was thrice as high at community-based testing sites than at PHC (63.5% vs. 21.0%; p = 0.001). The median age was similar in both community-based testing sites and PHCs (p = 0.389).

### HIVST: HIV self-tests; IQR: interquartile range

Characteristics of PRs by study site are shown in (**Table 2**). Most PRs (6,986; 70.4%) were female. Compared to PHCs, community-based testing sites had significantly more male PRs (51.3% vs 18.6%; p<0.001), and significantly more PRs who reported their HIV status as unknown (43.9% vs 30.0%; p<0.001).

### Characteristics of individuals tested for HIV (HIVST users and non-users)

During the study period, 34,715 adults were tested for HIV at both PHCs and community-based testing sites, among whom 1,089 (3.1%) participants reported HIVST use. Among

**Table 1. Acceptability of HIV Self-test at study sites.**

| Individuals approached | Community HIV Testing Sites (n = 5,181) | | | Primary Health Clinics (n = 31,527) | | | Total (n = 36,708) |
|---|---|---|---|---|---|---|---|
| Sex | Men | Women | Subtotal | Men | Women | Subtotal | |
| Acceptability, n (%) | 1,709 | 1,584 | 3,293(63.5%) | 1,264 | 5,334 | 6,598(21.0%) | 9,891 (26.9%) |
| HIVST kits distributed Median (IQR) | 3(2–5) | 3(2–4) | 3(2–4) | 3(2–4) | 3(2–4) | 3(2–4) | 3(2–4) |
| Number HIVST distributed | 10,643 (34%) | | | 20,698 (66%) | | | 31,341 (100%) |

**Table 2. Characteristics of primary recipients at distribution sites.**

| | Community Testing Sites (n = 3,293) | Primary health facilities (n = 6,598) | Total (N = 9,891) | p-value |
|---|---|---|---|---|
| **Sex n(%)** | | | | |
| Female | 1,604(48.7) | 5,334(81.4) | 6,986(70.6) | <**0.001** |
| Male | 1,689(51.3) | 1,216(18.6) | 2,905(29.4) | |
| **Age group n(%) years** | | | | |
| 18 to 25 | 1155(35.1) | 2,172(33.0) | 3,327(33.6) | 0.076 |
| >25 to ≤39 | 1,507(45.7) | 3,160(47.9) | 4,667(47.2) | |
| ≥40 | 631(19.2) | 1,266(19.1) | 1,897(19.2) | |
| **HIV status n(%)** | | | | |
| Known Positive | 152(4.6) | 812(12.3) | 964(9.7) | <**0.001** |
| Known Negative | 1,693(51.4) | 3,812(57.7) | 5,505(55.7) | |
| Unknown | 1,448(43.9) | 1974(30.0) | 3,422(34.6) | |
| **HIVST kits** | | | | |
| 2 kits | 1,361(41.3) | 2,977(45.1) | 4,338(43.9) | **0.001** |
| 3 kits | 782(23.7) | 1,456(22.1) | 2,238(22.6) | |
| 4 kits | 396(12.0) | 673(10.2) | 1,069(10.8) | |
| 5 kits | 754(22.9) | 1,492(22.6) | 2,246(22.7) | |

HIVST users, 196 (17.9%) of individuals were confirmed HIV positive, while 2,685 (8.0%) HIVST non-users were confirmed HIV positive (**Fig 1**).

Among those who reported HIVST use, 640 (58.8%) were PRs (first recipients of the test) and 449 (41.2%) reported being SRs (having received the test from a PR). Characteristics of HIVST users are shown in **Table 3**.

The ages of PRs and SRs were similar, median age of 28.1 years (IQR: 23.5–34.7) and 26.9 years (IQR: 22.4–34.4) respectively (p = 0.07). Overall, 893 (82.0%) reported a negative HIVST, 196 (18.0%) reported a positive result and no participant reported an indeterminate result. Among HIVST users, 12 (1.1%) reported never having been tested, whereas 76 (7%) had last tested for HIV more than a year ago and 870 (79.9%) tested within the last six months. Through HIVST, HIV positivity was more than 10-fold higher in PHC than in community-based testing sites (3.3% vs 36.1%; p<0.001).

## Return for confirmatory testing and ART uptake

Of 196 (18%) who reported positive HIVST results and arrived at both PHCs and community-based testing sites for confirmatory testing, all reported having performed the HIVST within a median time of two months (IQR: 1.15–2.85). Of 196 (100%) confirmed HIV positive through HTS, clinical records of 36 (18.4%) participants could not be found after extensive search of electronic registers for the study sites (**Fig 1**). These were categorized as not linked to care. In addition, 25/160 (15.6%) were already on ART before receiving HIVST with median time in care of 53.2 months (IQR: 32.8–68.6), 129/171 (75.4%) initiated ART, whereas 2,362/2685 (88%) of HIV positive HIVST non-users-initiated ART.

## 4. Discussion

Distribution of HIVST kits to persons attending primary care clinics or community-based testing sites for self-testing and/or distribution to peer was acceptable. More than a quarter of the people visiting clinics and community-based testing sites accepted to be PRs, with 31,341 HIVST kits distributed within 17 months of implementation. Although the majority of HIVST

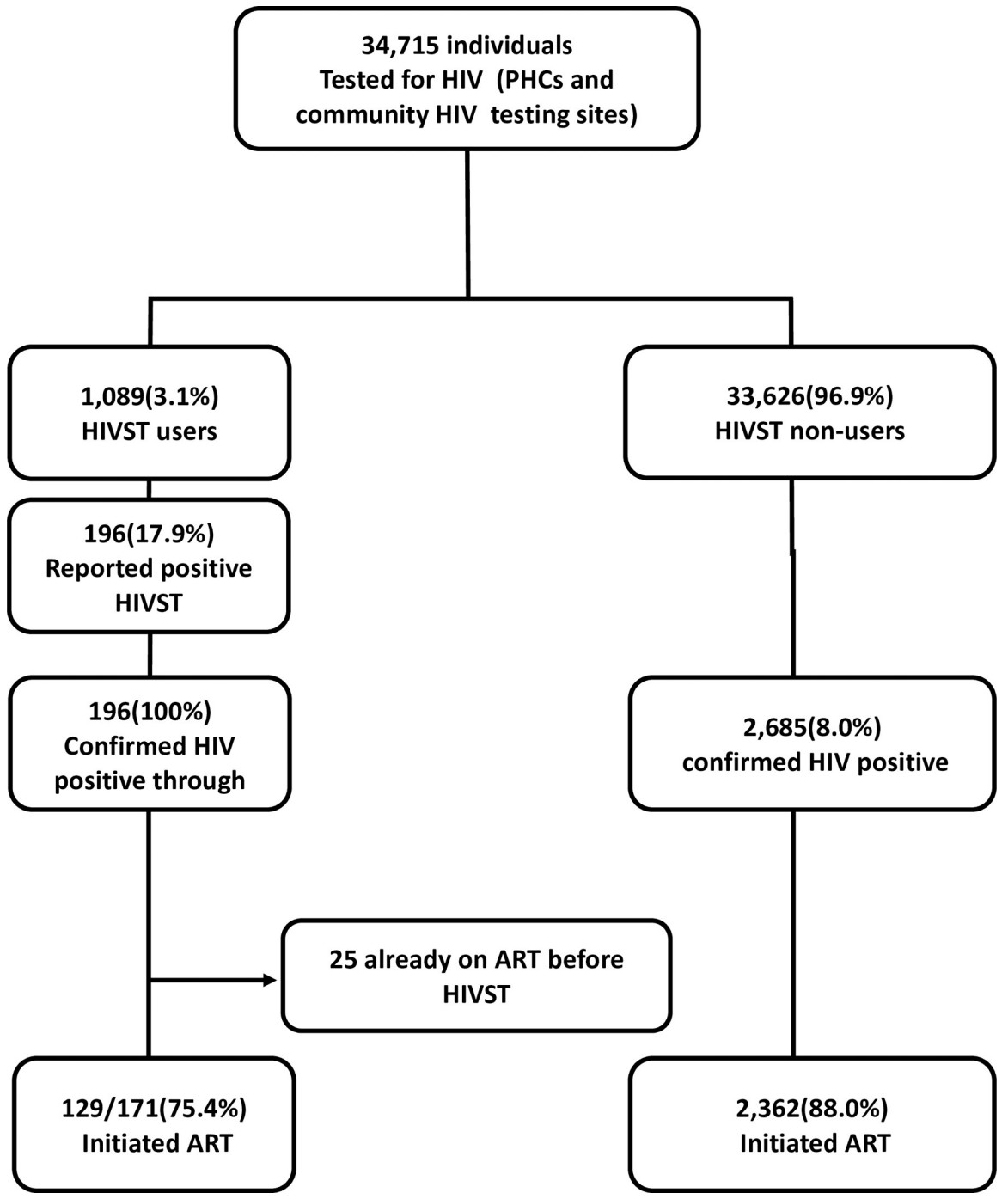

**Fig 1. Flow diagram for passive follow up component.**

kits (66%) were distributed in PHCs, which is due to their higher footfall, acceptability was thrice as high in community-based testing sites (63.5%) compared to clinics (20.9%). This could be explained by the fact that CHWs in community-based testing sites were able to spend more time with people, explaining the use of HIVST than HIV counsellors at busy PHCs. Studies conducted in Sub-Saharan African (SSA) countries evaluating the acceptability of HIVST among individuals to act as PRs have shown higher acceptability rates ranging from

**Table 3. Characteristics of HIVST users coming to the facility to perform HIV test.**

| | Community testing sites (n = 601) | Primary health facilities (n = 488) | All sites (n = 1089) |
|---|---|---|---|
| **Participants type n(%)** | | | |
| Primary recipients | 377(62.7) | 263(53.9) | 640(58.8) |
| Secondary recipients | 224(37.3) | 255(46.1) | 449(41.2) |
| **Sex n(%)** | | | |
| Male | 218(36.3) | 154(31.6) | 372(34.2) |
| Female | 383(63.7) | 334(68.4) | 717(65.8) |
| **Age group n(%)yrs** | | | |
| 18 to 25 | 210(35.0) | 174(35.6) | 384(35.3) |
| >25 to ≤40 | 322(53.5) | 236(48.4) | 558(51.2) |
| ≥40 | 69(11.5) | 78(16.0) | 147(13.5) |
| **HIVST Results n(%)** | | | |
| Positive | 20(3.3) | 176(36.1) | 196(18.0) |
| Negative | 581(96.7) | 312(63.9) | 893(82.0) |
| **HCT Results n(%)** | | | |
| Positive | 23(3.8) | 172(35.2) | 195(18.0) |
| Negative | 578(96.2) | 316(64.8) | 894(82.0) |
| **Testing history n(%)** | | | |
| Never tested | 1(0.2) | 11(2.2) | 12(1.1) |
| Tested >1 yr ago | 58(9.6) | 18(3.7) | 76(7.0) |
| Tested 6m-1 yr ago | 91(15.1) | 24(4.9) | 115(10.6) |
| <6mth ago | 450(74.9) | 420(86.1) | 870(79.9) |
| Not reported | 1(0.2) | 15(3.1) | 16(1.4) |

31% to 95% [20, 23–26]. However, these studies had primarily focused on either pregnant woman, post-partum women or female sex workers, had used monetary incentives, and were conducted under controlled conditions. Our finding suggests making HIVST more accessible at community-based testing sites might help to improve testing uptake and HIV diagnosis in SSA.

Individuals receiving a positive self-test result are recommended to seek confirmatory testing for the diagnosis of HIV [8]. During the study period, 1,089 individuals reported HIVST use; of these 58.8% tested themselves. Thus, not only did people agree to distribute HIVST, but they also used it as a method to test themselves in private. This can help, along with reaching more people, also to increase HIV testing frequency. Furthermore, a higher proportion of HIVST users had tested less six months before and a smaller proportion were first-time testers. This is not surprising considering the high HIV testing coverage in the study area [22]. Further research is needed on how HIVST will co-exist within HTS and on the role of HIVST in settings with high HIV testing coverage.

Linkage to confirmatory testing and HIV care after a positive HIVST has been highlighted as a major concern in literature from SSA given that not many people have the motivation to seek confirmatory HIV testing and/or linkage to care following an HIV- positive self-tested result [27]. In this study a large of people who returned for confirmatory testing reported a negative HIVST result, despite clear instruction that they should not return for confirmatory testing following a negative HIVST result. Further research is needed to explore reasons which lead those with HIVST negative results to come back for confirmatory testing.

We were unable to measure the true linkage to confirmatory testing given that we did not actively follow-up participants in order ascertain their HIVST results. Estimating linkage

following positive HIVST result is complex, and there is no single measurement strategy that will fit the needs of all HIVST implementers and researchers [13]. However, studies that successfully estimated linkage to confirmatory testing and HIV care reported a higher linkage to confirmatory testing [17, 18]. A cohort study in Kenya that actively promoted linkage to care, reported that linkage to care following HIVST can be comparable to the national proportion of linkage under universal test and treat policy [26]. Further research is needed to identify innovative ways to estimate linkage to confirmatory testing following HIVST and the amount of time required to link, particularly under programmatic conditions.

The availability of electronic register (tier.net) containing retrospective CD4 and vital load allowed important insights into awareness of positive status of HIVST users. Over 10% of those confirmed HIV positives had been diagnosed prior to study enrolment, despite clear instruction that the test should not be used by PLHIV on ART and waning of false positive HIVST if on ART [6, 28]. Repeat testing by those on ART has been reported elsewhere [29, 30] and suggests that PLHIV desire to re-confirm their HIV status for themselves, despite their reluctance to disclose their status. Programs should spend more time on patient education so that those on ART must no use HIVST.

An important contribution of this study was an attempt to establish the proportion of individuals who returned for confirmatory testing and ART uptake following HIVST. In our study, confirmatory testing and ART uptake was lower among HIVST users compared to HIVST non-user, 75.4% and 88.0% respectively. The universal test and treat (UTT) policy might have contributed to this difference. UTT interventions are intensive testing programmes designed to find even the hardest-to-reach populations. Systematic review and meta-analysis of linkage to treatment or care following HIVST in the general population has found limited but encouraging evidence of successful linkage among HIVST users [31]. Further research is needed to assess cost-effective and sustainable interventions designed to strengthen linkage to treatment and care following HIVST, such as assisted referrals and home ART initiation. Additional data is needed to assess the effect on HIVST on linkage to prevention services among those who are HIV-negative and at risk under, particularly in HIV-high prevalence and low-income rural settings.

## Strengths and limitations

To the best of our knowledge, this study constitutes the first study using unsupervised peer-distribution of an HIVST to people attending primary care clinics or community-based testing sites under programmatic conditions in a high HIV prevalence context, characterized by poor infrastructure and shortage of staff. Regular, public sector clinic staff at the study sites implemented the study, demonstrating how HIVST peer-distribution strategy would work in the real-word conditions at public health facilities and community, therefore maximizing generalizability.

This study has several limitations that warrant discussion. Firstly, the rural and semi-urban settings may limit its generalizability to urban settings. The use of routine data to assess the initiation of ART limits reliability of this variable. An inherent limitation of many studies of unassisted self-testing is the difficulty of verifying use and accuracy of the tests. In our study all information in the follow-up component was obtained through participants' reports, and thus we were unable to verify use, or results. However, PRs were told that the HIVST kits received were for peer-distribution or self-testing and reporting bias is likely to have been minimal. The fact that 640/1089 (58.8%) people during follow-up reported having tested-themselves, this supports the notion of limited reporting bias.

We had planned for a period of six months following the stoppage of HIVST distribution, during which people who took longer would have linked to confirmatory testing. However,

this period had to be shortened to three months due to COVID-19 pandemic. Hence, we might have missed some HIVST returning users, resulting in under-estimation of linkage to confirmatory testing and initiation of ART.

## 5. Conclusion

The WHO and UNAIDS had previously emphasized the need to develop a larger evidence base on HIVST to better inform national guidelines and implementation of HIVST services. This study responds to this need. Unsupervised peer-distribution of HIVST to people visiting primary care clinics or community-based testing sites was acceptable, with more than 1 out 4 accepting to be peer-distributors. Acceptability to be PRs was thrice as high in community sites compared to clinics. However, few secondary recipients were first time testers, young adults, and men, indicating this distribution model did not reach the hard-to-reach population. The data also suggest the scale up of HIVST has the potential to increase linkage to care and initiation of ART.

## Acknowledgments

The authors thank the health authorities of the KwaZulu-Natal Department of Health, King Cetshwayo District, for providing access to health facilities and community-based testing sites to conduct the study. We thank the staff of Médecins sans Frontières, Eshowe Project and the staff of Shine for their contribution and support, particularly the counselors and community health workers who distributed HIVST, and conducted enrolment and follow-up. We thank all the participants for their willing participation.

## Author Contributions

**Conceptualization:** Marcel K. Kitenge, Chinmay Laxmeshwar, Elkin Bermudez Aza, Ellie Ford-Kamara, Ntombi Gcwensa, Esther C. Casas, Khanyo Hlophe, Petros Isaakidis, Liesbet Ohler.

**Data curation:** Marcel K. Kitenge, Chinmay Laxmeshwar, Khanyo Hlophe, Petros Isaakidis, Liesbet Ohler.

**Formal analysis:** Marcel K. Kitenge, Chinmay Laxmeshwar, Gilles Van Cutsem, Esther C. Casas, Khanyo Hlophe, Petros Isaakidis, Liesbet Ohler.

**Investigation:** Marcel K. Kitenge, Chinmay Laxmeshwar, Ellie Ford-Kamara, Gilles Van Cutsem, Ntombi Gcwensa, Esther C. Casas, Khanyo Hlophe, Petros Isaakidis, Liesbet Ohler.

**Methodology:** Marcel K. Kitenge, Elkin Bermudez Aza, Ellie Ford-Kamara, Gilles Van Cutsem, Ntombi Gcwensa, Esther C. Casas, Khanyo Hlophe, Petros Isaakidis, Liesbet Ohler.

**Project administration:** Marcel K. Kitenge, Petros Isaakidis.

**Resources:** Marcel K. Kitenge, Chinmay Laxmeshwar, Ntombi Gcwensa, Khanyo Hlophe, Petros Isaakidis, Liesbet Ohler.

**Software:** Marcel K. Kitenge, Chinmay Laxmeshwar, Esther C. Casas.

**Supervision:** Marcel K. Kitenge, Chinmay Laxmeshwar, Gilles Van Cutsem, Ntombi Gcwensa, Esther C. Casas, Khanyo Hlophe, Petros Isaakidis, Liesbet Ohler.

**Validation:** Marcel K. Kitenge, Chinmay Laxmeshwar, Elkin Bermudez Aza, Ellie Ford-Kamara, Gilles Van Cutsem, Ntombi Gcwensa, Esther C. Casas, Khanyo Hlophe, Petros Isaakidis, Liesbet Ohler.

**Visualization:** Marcel K. Kitenge, Chinmay Laxmeshwar, Elkin Bermudez Aza, Gilles Van Cutsem, Esther C. Casas, Khanyo Hlophe, Petros Isaakidis.

**Writing – original draft:** Marcel K. Kitenge.

**Writing – review & editing:** Marcel K. Kitenge, Chinmay Laxmeshwar, Elkin Bermudez Aza, Ellie Ford-Kamara, Gilles Van Cutsem, Ntombi Gcwensa, Esther C. Casas, Khanyo Hlophe, Petros Isaakidis, Liesbet Ohler.

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
