## [Decision Letter · Decision Letter 0]

28 Sep 2021

PONE-D-21-19545Feasibility and acceptability of unsupervised peer-based distribution of HIV oral self-testing for the hard-to-reach in rural KwaZulu Natal, South Africa: Results from a demonstration studyPLOS ONE

Dear Dr. Marcel Kanyindda Kitenge,

Thank you for submitting your manuscript to PLOS ONE. After careful consideration, we feel that it has merit but does not fully meet PLOS ONE’s publication criteria as it currently stands. Therefore, we invite you to submit a revised version of the manuscript that addresses the points raised during the review process.

We look forward to receiving your revised manuscript.

Kind regards,

Limakatso Lebina, MBChB

Academic Editor

PLOS ONE

Journal Requirements:

Additional Editor Comments (if provided):

We apologise for the delays in providing feedback. Reviewers requested additional time.

Reviewers' comments:

Reviewer's Responses to Questions

**Comments to the Author**

1. Is the manuscript technically sound, and do the data support the conclusions?

Reviewer #1: Partly

Reviewer #2: Yes

Reviewer #3: No

2. Has the statistical analysis been performed appropriately and rigorously? 

Reviewer #1: I Don't Know

Reviewer #2: Yes

Reviewer #3: No

3. Have the authors made all data underlying the findings in their manuscript fully available?

Reviewer #1: No

Reviewer #2: Yes

Reviewer #3: Yes

4. Is the manuscript presented in an intelligible fashion and written in standard English?

Reviewer #1: Yes

Reviewer #2: Yes

Reviewer #3: No

5. Review Comments to the Author

Reviewer #1: The author presented a fair background and literature review. However, the methodology section is poorly articulated and needs improvement. The design of the study is insufficiently described as far as scientific methodology design is concerned. Authors needs to address the following concerns

concerns:

Background

line 84:More than two-thirds of people living with HIV (PLHIV) are in Africa'' it would great to make reference of the year.

line 85: sentence seems incomplete

Methods

line 116: Expand your design section, read on scientific methodological design

line 148_149 ''Those diagnosed with HIV on the 148 day of visit were excluded'' what was the reason for excluding, would you agree that perhaps the partners of those whom tested positive might have needed the kits or missed due to this , and was everyone tested before handing out the HIVST kits.

Passive follow up:

why passive follow up?

What happened to those who reported back to not study/participating facilities. Were neighboring facilities informed of the ongoing study and how to respond to HIVST bought back?

Results :

line 221: 34,715 adults were tested for HIV at both PHCs and community-based testing sites, among whom 1,089 participants reported HIVST use'' so the 1089 already knew their HIV status before being handed the test kits?

line 223: the 196 individuals who tested positive, was this on site or during follow up, assuming everyone was tested, this means some of these ( PR's) were HIV negative at baseline? please clarify

Figure 1: Only HIV positives reported, what happened to the other HIVST, are we assuming they were all negative, where there any unused kits returned to the facilities or study sites?

Reviewer #2: The study assesses the acceptability of HIV self-testing (HIVST) in primary health care and community service delivery settings in areas of KwaZulu Natal, South Africa. HIVST kits were distributed through primary recipients to secondary recipients in their social networks over a 17-month period. Acceptability of HIVST was assessed along with ART initiation for participants who tested HIV-positive. Although acceptability of HIVST had been established by several studies, this study aimed at filling gaps in evidence acceptability of unsupervised peer distribution of HIVST kits in under-served primary health care and community settings. The study, based on the data presented, concluded that HIVST unsupervised peer-distribution of HIVST was acceptable, particularly at community sites and suggested that this has the potential to increase linkage to HIV care and treatment. The article presents sound and informative research output on HIVST, and has important implications for HIV programs and delivery of HIV testing services, particularly in resource-limited settings.

Authors may wish to consider the comments/suggestions below to improve the quality of the manuscript.

Do not see the reason why ‘feasibility’ is included in this manuscript. Feasibility (feasible) is mentioned in the title, in the background (as related to other studies which looked at feasibility), in the first sentence of the Discussion section(as a finding from the study), and in the Conclusion (only in the Abstract section). There isn’t anything related to this in the methods or results. May need to remove this (feasibility/feasible) from the title and discussion and conclusion.

Abstract: word count is 384 (vs 300 word limit for PLoS One). Many acronyms/ abbreviations are used (vs abstracts should not include Abbreviations, if possible, in PLoS ONE guideline)

Background: provides a good appraisal of what is known in the area and clearly highlights the rationale for conducting this study (and it is ‘acceptability’, nothing more mentioned here)

Methods: from what is presented in this section, the study used a (prospective) cross-section design. One may even consider this as a cross-sectional analytical design, as clearly stated in the Analysis section (Line 184…) “Pearson’s Chi-squared and Fisher’s exact tests were used to assess the statistical significance of differences in categorical outcomes between selected groups. Two-tailed tests were used throughout, and the level of significance was set at 5%”. This means the one may describe this study as a cross-sectional analytical design (with prospective data collection). Also, a sample size is required for this study. Of note, it is also stated in Line 321 “The study was not powered to detect factors associated with linkage confirmatory testing and initiation of ART. What is power for? --- Sample Size? By just eyeballing, the sample seems adequate, but can one slip through without showing the sample size and the assumption involved in the estimation?

Results: Table captions should be complete to make them self-contained (add to the titles given place and date. It is helpful to provide expanded forms of unusual acronyms in the titles and bodies of tables as footnotes to the tables. Descriptions of some of the results provided in the tables may need to move to more appropriate sections/paragraphs to improve on the flow of content (comments given in sections of the results below). Use commas consistently in figures more than 4 digits, like 1,000 instead of 1000).

Specific comments:

Abstract:

Line 38: There is an extra space between the hyphen and distribution

Line 42: May change partner to partners

Line 50: ‘accepted to be’, I think, should be changed to ‘were’

Line 52: ‘6,598 (66.7%)’ is out of place in the sentence; please edit.

Line 58: I think, “who’ is missing between ‘64.7%)’ and ‘were”

Line 66: For reasons given in #1 under general comments above, “feasible and” may be deleted.

Background:

Line 85: The sentence is not complete; something is missing before (1).

Line 85: There is an extra pace between 2030- and stating

Line 102: ‘services’ after (HTS) should be deleted

Line 109: Change hyphen to comma in the ref (18-19)

Line 112: I think, quite rightly include only ‘acceptability’

Methods:

Line 166: Maybe in order to replace descriptive with cross-sectional analytical.

Line 118: “areas’ – can one more be specific? -- like by administrative structure.

Line 133 (study definitions): It could be helpful to add here operational definitions for ‘acceptability’ and ‘hard to reach’.

Line 184: Need to add median to frequencies and proportions.

Line 189 (Ethical considerations): Not sure, but wondering whether one would need to add a few lines addressing some key ethical issues here like consent and confidentiality, etc.

Results:

Line 195: …. “PRs, of them 9,891(26.9%) accepted” may be edited as … 'PRs of which 9,891(26.9%) accepted.'

Table 1: Useful to add % men and % female for each group to show acceptability by sex.

Lines 207-209: This is a description of content Table 1, not in Table 2. Should move this to the description for Table 1.

LINE 208: 64.5% here vs 63,5 in Table 1.

Table 2: Age group class intervals >25 to ≤40 and ≥40 overlap; please correct. All P-values should be in the variables’ value rows.

Line 223: '196 individuals or 17.9% (95% CI 15.7 to 20.3)' – Confidence interval is not shown in the figure or any of the table. One may edit this as “..196 (17.9%) of the individuals ..”

Lines 221-224: Better to move this and include it in the description at the beginning of the section on 'Return for confirmatory testing and ART uptake' below, where Figure 2 =3 is presented.

Line 228: insert word median before 28.1 years.

Line 230. Remove the word 'furthermore' and just start the sentence with “Among”

Line 231: insert a comma after users.

Line 222-233: Remove 'In addition' and just start with 'Through'

Table 3: Remove the full stop at the end of the title. Age group class intervals >25 to ≤40 and ≥40 overlap; please correct.

Line 241. Add 'a median time of' before 'two months' and add (100%) after 190.

Line 242. Insert space between 36 and (18.4%).

Line 247: Delete 'Linkage to care and ART initiation'

Discussion:

Line 256: May need to delete 'and feasible' for reasons given above.

Line 263: SSA expanded here; the acronym should be used subsequently (lines 280, 385) – please edit.

Line 267: Replace 'This' with 'Our', otherwise one may This to mean the studies describe in the preceding sentence.

Line 271: Edit '1089' as '1,089'

Line 330 (Conclusion): there is more in the conclusion of the Abstract than here. May need to add

Some more here.

Acknowledgment:

Line 343: Add 'and' between HIVST, and conducted

Availability of data and materials

Please see the Journal guidelines where to include this.

Funding

Please see the Journal guidelines where to include this.

Reviewer #3: This paper presents a model to distribute oral HIV self-tests (HIVST) in rural South Africa and describes the acceptability and feasibility of unsupervised peer distribution of HIV self-testing. This paper offers an a potentially important model for reaching hard to reach rural populations with HIV testing; however, there are a number of issues with the paper in its current form that warrant further attention. In particular:

• The description of previous studies of HIVST in lines 101-103 is not clear. I suggest the authors state more clearly what these previous studies found and how the current study adds to this literature.

• The “Design” section of the Methods (lines 115-116) does not describe the evaluation design of this study.

• The “Study definitions” section (lines 133-139) seems like it would be better labeled “participants”.

• The authors note in lines 148-149 that persons diagnosed with HIV were excluded from participation. Can the authors explain why that was, given that one of the purposes of distributing HIVST kits is to offer sexual partners an opportunity to test?

• In the “Data analysis” section (lines 182-184), the authors define “acceptability” as “the proportion of people invited to be PRs who accepted HIVST”. This is a bit confusing to me – is acceptability the number of people who agreed to take the tests and distribute them to others or is it the number of people who agreed to use the self-test?

• Acceptability is typically defined as uptake – that is – are people willing to use the test. It has been pretty well established in the literature (including what is cited in this paper) that HIV self-testing is acceptable to most people in South Africa. I think the potential contribution of this paper is on the effectiveness (or not) of the distribution model for hard-to-reach populations in rural settings. That is, did it improve testing access or coverage in community and/or clinics settings? What evidence do the authors have to show that and how did it improve access and/or coverage specifically within the clinics (or were the kits taken outside of the clinic for broader distribution?) And beyond the community testing site? More on the how would be helpful.

• In Table 1 and in the description of the data (lines 195-201), it seems that the authors are describing acceptability as the number of people who were willing to take the HIVST kits and distribute them – is this correct? Also, the authors report on acceptability again in lines 207-208 showing a difference between community and primary health clinic sites. This is a bit confusing as most people think of this as how many people actually took the HIVST kit to test/use rather than the distributors. Is there a way to label this differently so it is clearer to the reader what the authors are describing and why that is important (e.g., the characteristics of the distributors?) or is this describing the characteristics of the people who actually tested with the HIVST kits?

• The authors report in lines 221-227 that of 34,715 adults that were tested for HIV at both primary care clinics and community-based testing sites during the study period, only 1,089 (3%) reported testing with an HIVST. This number is quite low and suggests potentially low coverage of distribution of the >31,000 HIVST kits or potential low uptake/use of the kits given the prevalence in the region and undiagnosed persons with HIV. The authors should consider why the number overall is low and discuss potential reasons for this in the Discussion section.

• In this same section, the authors report that the majority of the reported users of the HIVST kits were the distributors (primary recipients – 59%) who had mostly been tested within the past 6 months. This begs the question then of how this model is improving access to testing and coverage since most of these folks may likely be retesting.

• The data in Figure 1 are a bit misleading. Specifically, in the column on the left it seems that the 36 confirmed positive persons whose records could not be found should not be removed from the denominator in the last box as these were confirmed positives but did not link or initiate ART. Similarly, the 25 already on ART seem to belong in the denominator as well since they are on ART. If the 25 are added back in, the number initiated or on ART would be 154/196 confirmed positives (79%) with 36 having no record of initiating ART and 6? This is a more accurate estimate of the number of PLHIV who initiated or were on treatment among the 196 HIVST PLHIV identified. The current estimate is an overestimate.

• As shown above, the HIVST users actually had lower treatment initiation or on ART rates than those tested through other methods. How to address this issue and strengthen linkage to care for HIVSTers should be addressed in the Discussion section.

• The Discussion section often repeats the results. Suggest that the authors briefly summarize the key findings without restating the data and offer information to help interpret the finding or discuss how it supports or contradicts the extant literature.

• Overall, this study has potential to propose a model for distribution of HIVST kits for use by hard-to-reach populations in rural settings in South Africa. However, there are current limitations in the way the data is presented and the lack of clarity of the findings that limit the interpretation of the data and its current contribution to the literature. The authors are encouraged to present the data in a way that would clearly show who is being reached with these test kits and if they are truly increasing access to testing in these populations (especially men and young people which appear to account for about a third of those using HIVST kits) or if they are, for example, primarily providing retesting options for those who are already testing regularly every 6 months.

• The paper needs a copyediting review before it is re-submitted as there are a number of missing words, spaces, and punctuation.

6. PLOS authors have the option to publish the peer review history of their article (what does this mean?). If published, this will include your full peer review and any attached files.

Reviewer #1: No

Reviewer #2: **Yes: **Sileshi Lulseged

Reviewer #3: No

---

## [Author Response · Author response to Decision Letter 0]

27 Jan 2022

To : The Editor

PLoS ONE 

Re: Responses to reviewers’ comments

Dear Editor,

We thank you and the reviewers for the comments. Please find below a point-by-point response to the issues raised by the reviewers. The changes made to the manuscript are highlighted in track changes in the revised version.

Original Title: Feasibility and acceptability of unsupervised peer-based distribution of HIV oral self-testing for the hard-to-reach in rural KwaZulu Natal, South Africa: Results from a demonstration study .

Reviewer #1 

Comment 1 

Background

line 84:More than two-thirds of people living with HIV (PLHIV) are in Africa'' it would great to make reference of the year.

Response

We thank the reviewer for the comment. We have added the year as shown in line 93 and it now read as follows “In 2019, more than two-thirds of people living with HIV (PLHIV) were in Africa.”

Comment 2

Methods

line 116: Expand your design section, read on scientific methodological design

Response 

We appreciate the reviewer’s comment. We have expanded the study design as shown 134-138 and this reads as follows “This was an analytical cross-sectional study of peer-led, unsupervised distribution model of HIVST kits. Whereby, lay counsellors or community health workers provided HIVST kits to primary recipients (PRs) for distribution to their sexual partner, anyone in their social network (termed secondary recipients) or for self-testing.”

Comment 3

line 148_149 ''Those diagnosed with HIV on the 148 day of visit were excluded'' what was the reason for excluding, would you agree that perhaps the partners of those whom tested positive might have needed the kits or missed due to this , and was everyone tested before handing out the HIVST kits.

Response 

We thank the reviewer for the comment. We fully agree with reviewer that excluding those whom tested positive on the day of the visit may have missed an opportunity to get their partners tested. The decision to exclude those testing HIV-positive on the day of the visit was motivated due to the consideration of the mental trauma that the diagnosis of HIV might lead to. The study team felt that given the psychological impact of a HIV-positive diagnosis, it might be not prudent to put the added onus of asking them to take HIVST kits to their partners, which might also need them to disclose their status. The study team also felt that the consent obtained from the newly diagnosed participant might not be out of free will and additionally the consent process might lead to additional unnecessary burden. Looking the complexities involved, it was decided to exclude those who tested HIV-positive on the day of the visit. They could be included in the study during one of their follow-up visits, so they would not miss out on the opportunity to access HIVST kits.

We have added justification of this exclusion in line 184-186 and this reads as follows “Given the emotional and psychological distress associated with a new HIV-positive diagnosis, those diagnosed with HIV on the day of visit were excluded.”

Comment 4

Passive follow up:

Why passive follow up?

What happened to those who reported back to not study/participating facilities. Were neighbouring facilities informed of the ongoing study and how to respond to HIVST bought back?

Response 

We appreciate the reviewer’s comment. The study used passive follow-up to demonstrate how HIVST peer-distribution strategy might work as part of routine programmatic setting where active follow-up might not be achieved for all. Therefore, actively following primary recipients would have defeated the intended objective of the study which was to demonstrate how unsupervised peer-distribution might work in programmatic setting. We have stated in line 200-203 that “After handing over the tests to the PRs, we did not actively follow-up with them. HIVST users would decide on their own agency to visit the clinic for confirmatory testing and ART initiation.” 

The study setting consisted of 27 sites (PHC and community-HIV testing sites). These were all sites serving the study area. During the study period, everyone coming to the public health facilities and community HIV testing sites were asked if they had used an HIVST and from whom did they receive an HIVST kit. As stated in line 205-208 and this reads as follows “During the study period, everyone coming to the study sites for HIV testing was asked if they had used HIVST and from whom they received an HIVST kit. In case they reported using HIVST, with or without presenting the care card, they were informed about the study and asked for consent to be enrolled in the study.” However, we do agree that some individuals may have reported outside of our catchment area. We had conducted an outreach to those healthcare sites close by, but outside our catchment areas and had requested them to ask such participants to report to one of the study sites. However, it cannot be denied that some participants might have reported to a facility that was not part of the study. We feel that this makes the study results representative of how the results of introducing HIVST in routine programmatic setting might look like. 

Comment 5

Results :

line 221: 34,715 adults were tested for HIV at both PHCs and community-based testing sites, among whom 1,089 participants reported HIVST use'' so the 1089 already knew their HIV status before being handed the test kits?

Response 

We thank the reviewer for the comment. As stated in previous response, during the study period, everyone coming to public health facilities in the catchment area was asked if they had used an HIVST and from whom they received an HIVST. During the study 34,715 adults were tested for HIV at both PHCs and community-based testing sites. Among them, 1089 reported having used an HIVST and had come for confirmatory testing. These included a mix of those who had collected the HIVST kit from the facility or had received the test kit from someone else who had come to the facility.

Comment 

line 223: the 196 individuals who tested positive, was this on site or during follow up, assuming everyone was tested, this means some of these ( PR's) were HIV negative at baseline? please clarify

 Response

All 196 who reported testing positive on HIVST received confirmatory testing on site during the follow-up. In lines 315-317 we mention, “Of 196 (18%) who reported positive HIVST results and arrived at both PHCs and community-based testing sites for confirmatory testing, all reported having performed the HIVST within a median time of two months (IQR: 1.15-2.85)”. Additionally, our study did not seek to actively follow-up PRs and we cannot confirm with certainty if some PRs had sero-converted during the study period. We have acknowledged this as one of the study limitations in line 499-402 and this reads as follows “An inherent limitation of many studies of unassisted self-testing is the difficulty of verifying use and accuracy of the tests. In our study all information in the follow-up component was obtained through participants’ reports, and thus we were unable to verify use, or results.”

Comment 

Figure 1: Only HIV positives reported, what happened to the other HIVST, are we assuming they were all negative, where there any unused kits returned to the facilities or study sites?

Response 

We thank the reviewer for the comment. Figure 1 aimed to graphically present linkage to care among HIVST users and non-users. This is shown in 299-300 and it reads as follows “Overall, 893 (82.0%) reported a negative HIVST, 196 (18.0%) reported a positive result and no participant reported an indeterminate result.” Additionally, this information is also included in Table 3. No unused kits were returned to the facilities or study sites. 

Reviewer #2 

Comment 

Do not see the reason why ‘feasibility’ is included in this manuscript. Feasibility (feasible) is mentioned in the title, in the background (as related to other studies which looked at feasibility), in the first sentence of the Discussion section(as a finding from the study), and in the Conclusion (only in the Abstract section). There isn’t anything related to this in the methods or results. May need to remove this (feasibility/feasible) from the title and discussion and conclusion

Response 

We thank the reviewer and agree with this suggestion. We have removed the words (feasibility/feasible) throughout the manuscript.

Comment

Abstract: word count is 384 (vs 300 word limit for PLoS One). Many acronyms/ abbreviations are used (vs abstracts should not include Abbreviations, if possible, in PLoS ONE guideline)

Response 

We appreciate the reviewer’s comment. We have edited the abstract to comply with the journal guidelines. 

Comment 

Background: provides a good appraisal of what is known in the area and clearly highlights the rationale for conducting this study (and it is ‘acceptability’, nothing more mentioned here)

Response 

We thank the reviewer for the kind words. As highlighted in our first response, we have deleted ‘feasibility’ or ‘feasible’ as you suggested in your first comment. 

Comment 

Methods: from what is presented in this section, the study used a (prospective) cross-section design. One may even consider this as a cross-sectional analytical design, as clearly stated in the Analysis section (Line 184…) “Pearson’s Chi-squared and Fisher’s exact tests were used to assess the statistical significance of differences in categorical outcomes between selected groups. Two-tailed tests were used throughout, and the level of significance was set at 5%”. This means the one may describe this study as a cross-sectional analytical design (with prospective data collection). Also, a sample size is required for this study. Of note, it is also stated in Line 321 “The study was not powered to detect factors associated with linkage confirmatory testing and initiation of ART. What is power for? --- Sample Size? By just eyeballing, the sample seems adequate, but can one slip through without showing the sample size and the assumption involved in the estimation?

Response 

 We thank the reviewer for the comment and suggestions. We have changed the study design as shown in line 134-138 and this reads as follows “This was an analytical cross-sectional study of peer-led, unsupervised distribution model of HIVST kits, whereby, lay counsellors or community health workers provided HIVST kits to primary recipients (PRs) for distribution to their sexual partner, anyone in their social network (termed secondary recipients) or for self-testing. In addition, data was collected prospectively among those returned for confirmatory testing”. 

Furthermore, in this study we distributed 31,341 HIVST kites through primary recipients in 17 month. One primary recipient could have received 2-5 HIVST kits for distribution and for use. Our sample size was based on operational fasibility rather than any statistical calculations. Therefore , we did not have fixed sample size in terms of number of primary recipients to enrol in the study. We did have fixed number of HIVST kits to give out. In order to avoid confusion we have deleted lines 321-324. 

Comment 

Results: Table captions should be complete to make them self-contained (add to the titles given place and date. It is helpful to provide expanded forms of unusual acronyms in the titles and bodies of tables as footnotes to the tables. Descriptions of some of the results provided in the tables may need to move to more appropriate sections/paragraphs to improve on the flow of content (comments given in sections of the results below). Use commas consistently in figures more than 4 digits, like 1,000 instead of 1000).

Response 

We thank the reviewer for the comments and suggestions. We have expanded forms of unusual acronyms as shown in line 255-261, we have inserted commas in figures with more than 4 digits. 

Comment 

Specific comments:

Abstract:

Line 38: There is an extra space between the hyphen and distribution

Response 

We appreciate the reviewer comment and this has been addressed as shown in line 38 and this reads as follows “This study aimed to describe the acceptability of unsupervised peer-distribution of HIVST as a method to scale-up HIV testing.”

Comment 

Line 42: May change partner to partners

Response 

We thank the reviewer for the comment. This has been addressed as shown in line 42 and this reads as follows “…or community health workers provided HIVST kits to primary recipients (PRs) for distribution to their sexual partners,”

Comment

Line 50: ‘accepted to be’, I think, should be changed to ‘were’

Response 

We appreciate the reviewer’s comment. This has been addressed as shown in line 50 and this reads as follows “Among 36,708 people approached to be primary recipients, 9,891 (26.9%) accepted”

Comment 

Line 52: ‘6,598 (66.7%)’ is out of place in the sentence; please edit.

Response 

We appreciate the reviewer for the comment. “6,598 (66.7%)” has been deleted as show in line 52 and this reads as follows “PRs were predominately recruited at primary health clinics (PHCs)”

Comment 

Line 58: I think, “who’ is missing between ‘64.7%)’ and ‘were”

Response 

We thank the reviewer for the comment . The sentence has been changed as shown in line 56-58 and this reads as follows “Among HIVST users, 893 (82.0%) returned to the clinic for confirmatory testing after testing negative on HIVST; 196 (17.9%) were confirmed HIV positive following a positive HIVST”. 

Comment 

Line 66: For reasons given in #1 under general comments above, “feasible and” may be deleted.

Response 

We appreciate the reviewer’s comment, we have changed the manuscript title as shown in line 1-2 and this reads as follows “Acceptability of unsupervised peer-based distribution of HIV oral self-testing for the hard-to-reach in rural KwaZulu Natal, South Africa: Results from a demonstration study”

Comment 

Background:

Line 85: The sentence is not complete; something is missing before (1).

Response 

We thank the reviewer for the comment, the missing number has been added as shown in line 94 and this reads as follows “Worldwide, 19% of PLHIV did not know their status in 2019 [2], while in South Africa this figure was 8%”

Comment 

Line 85: There is an extra space between 2030- and stating

Response 

We thank the reviewer for the comment and the extra space has been removed as shown in line 96. 

Comment 

Line 102: ‘services’ after (HTS) should be deleted

Response

We thanks the reviewer for the comment, services’ after (HTS) has been deleted as shown in line 110-111 and this reads as follows “Evidence from South Africa has demonstrated that HIVST can be a feasible option to improve uptake of HIV testing services (HTS) at clinics”

Comment 

Line 109: Change hyphen to comma in the ref (18-19)

Response 

We thank the reviewer for the suggestion. Hyphen has been changed to comma and as shown in 127.

Comment 

Line 112: I think, quite rightly include only ‘acceptability’

Response 

We appreciate the reviewer’s comment and have edited the manuscript to include only acceptability.

Comment 

Methods:

Line 166: Maybe in order to replace descriptive with cross-sectional analytical.

Response 

We thank the reviewer for the comment. Descriptive has been changes to cross sectional analytical study. This as shown in line 134-138 and this reads as follows “This was a cross-sectional analytical study of peer-led, unsupervised distribution model of HIVST kits, whereby, lay counsellors or community health workers provided HIVST kits to primary recipients (PRs) for distribution to their sexual partner, anyone in their social network (termed secondary recipients) or for self-testing.”

Comment 

Line 118: “areas’ – can one more be specific? -- like by administrative structure.

Response 

We appreciate the reviewer’s comment. We changed areas to area shown in line 140 and this reads as follows “Eshowe and Mbongolwane areas in KwaZulu-Natal”. 

Comment 

Line 133 (study definitions): It could be helpful to add here operational definitions for ‘acceptability’ and ‘hard to reach’.

Response 

We appreciate the reviewer for the comment. Operational definition for the hard-to-reach has been added as shown in line 163-165 and this reads as follows “Hard-to-reach groups include sub-groups of the population that may be difficult to reach through conventional HIV testing services and include those who have no easy access to testing facilities and men, adolescents and young adults.”. Furthermore, operational definition for “acceptability” is found in line 172-174 and this reads as follows “Acceptability was defined as the proportion of people invited to be PRs and agreed to take HIVST kits for use and distribute them to their sexual partner, family members or anyone in their social network.” 

Comment

Line 184: Need to add median to frequencies and proportions.

Response 

We thank the reviewer for the comment. Data analysis section has been changed as shown in line 222-224 and this reads as follows “Continuous variables were summarised using mean and standard deviation if normally distributed, otherwise median and interquartile range were used, and categorical variables were summarised using percentages or proportions.”

Comment 

Line 189 (Ethical considerations): Not sure, but wondering whether one would need to add a few lines addressing some key ethical issues here like consent and confidentiality, etc.

Response 

We appreciate the reviewer for the comment. We have expanded ethical considerations section as shown in line 238-241 and this reads as follows “All PRs and SRs provided written informed consents before undergoing the study procedures and information such as age, gender, number of tests were collected. We did not collect any identifying information such as name, surname, address, phone number, or ID numbers.”

Comment 

Results:

Line 195: …. “PRs, of them 9,891(26.9%) accepted” may be edited as … 'PRs of which 9,891(26.9%) accepted.'

Table 1: Useful to add % men and % female for each group to show acceptability by sex.

Response 

We appreciate the reviewer’s comment. We have addressed the comment as shown in line 244-246 and this reds as follows “Between September 2018 and March 2020, 36,708 individuals were approached to participate in the study as PRs, of among whom 9,891(26.9%) accepted. Among the 9,891 who accepted, 2,973(30.0%) were men and 6,918(70.0%) were female.”

Comment 

Lines 207-209: This is a description of content Table 1, not in Table 2. Should move this to the description for Table 1.

Response 

We thank the reviewer for the suggestion. The section was moved as suggested as shown in line 250-252 and this reads as follows “Acceptability of HIVST was thrice as high at community-based testing sites than at PHC (64.5% vs. 21.0%; p=0.001). The median age was similar in both community-based testing sites and PHCs (p=0.389).”

Comment 

LINE 208: 64.5% here vs 63,5 in Table 1.

Table 2: Age group class intervals >25 to ≤40 and ≥40 overlap; please correct. All P-values should be in the variables’ value rows.

Response 

We appreciate the reviewer for the comment. The inconsistency between number in the text and the table has been fixed and this is shown in line 250-252 and it reads as follows “Acceptability was thrice as high at community-based testing sites than at PHC (63.5% vs. 21.0%; p=0.001)”. Additionally, we have corrected the age group class intervals >25 to ≤40 and ≥40 overlap as well as all P-values and these corrections are shown in table 2 . 

Comment 

Line 223: '196 individuals or 17.9% (95% CI 15.7 to 20.3)' – Confidence interval is not shown in the figure or any of the table. One may edit this as “..196 (17.9%) of the individuals .”

Response 

We thank the reviewer for the comment. This has been changed as shown in line 287 and reads as follows “…196 (17.9%) of individuals were confirmed HIV positive”. 

Comment 

Lines 221-224: Better to move this and include it in the description at the beginning of the section on 'Return for confirmatory testing and ART uptake' below, where Figure 2 =3 is presented.

Response 

we thank the reviewer’s suggestion. We moved Figure 1 to align this with the text as shown 289-293. 

Comment 

Line 228: insert word median before 28.1 years.

Response 

We appreciate the reviewer for the suggestion. We have inserted the word median as line 298 and this reads as follows “The ages of PRs and SRs were similar, median age of 28.1 years (IQR: 23.5-34.7)…”

Comment 

Line 230. Remove the word 'furthermore' and just start the sentence with “Among”

Response 

We appreciate the reviewer for the suggestion. The word ‘furthermore’ has been deleted as shown in line 308 and this reds as follows “Among HIVST users, 12 (1.1%) reported never having been tested,…”

Comment

Line 231: insert a comma after users.

Response 

We thank the reviewer for the suggestion. Comma has been inserted as suggested as shown in line 308 and this reads as follows “Among HIVST users, 12 (1.1%) reported never having been tested,…”

Comment 

Line 222-233: Remove 'In addition' and just start with 'Through'

Response 

We thank the reviewer for the suggestion. We have removed ‘In addition’ as suggested , this is shown in line 309-311 and this reads as follows “Through HIVST, HIV positivity was more than 10-fold higher in PHC than in community-based testing sites…”

Comment 

Table 3: Remove the full stop at the end of the title. Age group class intervals >25 to ≤40 and ≥40 overlap; please correct.

Response 

We appreciate the reviewer’s comment. Comma has been removed from the tile as show in table 3. 

Comment 

Line 241. Add 'a median time of' before 'two months' and add (100%) after 190 and Line 242. Insert space between 36 and (18.4%).

Response 

We appreciate the reviewer for the suggestions. We have added these as shown in line 316-323 and these read as follow “…all reported having performed the HIVST within a median time two months (IQR: 1.15-2.85). Of 196 (100%) confirmed HIV positive through HTS, clinical records of 36 (18.4%) participants could not be found after extensive search of electronic registers for the study sites (Figure 1).”

Comment 

Line 247: Delete 'Linkage to care and ART initiation'

Response 

We appreciate the reviewer’s comment, this has been deleted. 

Comment 

Discussion:

Line 256: May need to delete 'and feasible' for reasons given above.

Response 

The word ‘feasible’ has been deleted as shown in line 328-329 and this reads as follows “Distribution of HIVST kits to persons attending primary care clinics or community-based testing sites for self-testing and/or distribution to peer was acceptable”. 

Comment 

Line 263: SSA expanded here; the acronym should be used subsequently (lines 280, 385) – please edit.

Response 

We thank the reviewer for the comment. The acronym has been used as suggested, this is shown line 356-359 and it reads as follows ‘Linkage to confirmatory testing and HIV care after a positive HIVST has been highlighted as a major concern in literature from SSA given that not many people have the motivation to seek confirmatory HIV testing and/or linkage to care following an HIV- positive self-tested result’

Comment 

Line 267: Replace 'This' with 'Our', otherwise one may This to mean the studies describe in the preceding sentence.

Response 

We thank the reviewer for the comment. we have replaced ‘This’ with ‘Our’ as shown in line 340 and this reads as follows “Our finding suggests making HIVST more accessible..”

Comment 

Line 271: Edit '1089' as '1,089'

Response 

We appreciate the reviewer’s suggestion. This has been edited accordingly as suggested as shown in line 344-345 and it read as follows “During the study period, 1,089 individuals reported HIVST use; of these 58.8% tested themselves”. 

Comment 

Line 330 (Conclusion): there is more in the conclusion of the Abstract than here. May need to add some more here.

Response 

We thank the reviewer for the comment. We have added a sentence in the conclusion as shown in line 422-430 and this reads as follows “Acceptability to be PRs was thrice as high in community sites compared to clinics. However, few of those secondary recipients who reported were first time testers, young adults, and men, indicating this distribution model did not reach the hard-to-reach population.”

Comment 

Acknowledgment:

Line 343: Add 'and' between HIVST, and conducted

Response 

We appreciate the reviewer’s comment and this has been addressed accordingly as shown in line 449. 

Comment

Availability of data and materials

Please see the Journal guidelines where to include this.

Response 

We thank the reviewer for the comment, we have moved the section accordingly and this shown in line 438-442. 

Reviewer #3

This paper presents a model to distribute oral HIV self-tests (HIVST) in rural South Africa and describes the acceptability and feasibility of unsupervised peer distribution of HIV self-testing. This paper offers an a potentially important model for reaching hard to reach rural populations with HIV testing; however, there are a number of issues with the paper in its current form that warrant further attention. In particular:

Comment 

The description of previous studies of HIVST in lines 110-113 is not clear. I suggest the authors state more clearly what these previous studies found and how the current study adds to this literature.

Response 

We appreciate the reviewer’s comment. We have rephrased the paragraph as shown in line 110-114 and this reads as follows “Evidence from South Africa has demonstrated that HIVST can be a feasible option to improve uptake of HIV testing services (HTS) and the use of HIVST in an unsupervised environment can be feasible and the results obtained from such testing can be reliable. However, linkage to care and confirmatory testing for the unsupervised HIVST and further care were relatively low”. 

Comment 

The “Design” section of the Methods (lines 115-116) does not describe the evaluation design of this study.

Response 

We thank the reviewer for the comment. We have expanded the study design section and shown in line 134-138 and this reads as follows “This was a cross-sectional analytical study of peer-led, unsupervised distribution model of HIVST kits, whereby, lay counsellors or community health workers provided HIVST kits to primary recipients (PRs) for distribution to their sexual partner, anyone in their social network (termed secondary recipients) or for self-testing. Data was collected prospectively among those returned for confirmatory testing”.

Comment 

The “Study definitions” section (lines 133-139) seems like it would be better labelled “participants”.

Response 

We appreciate the reviewer’s comment. We added additional operational definitions to this section unrelated to the study participant and this is shown in line 162-174. Therefore, we maintain the current wording. 

Comment 

The authors note in lines 148-149 that persons diagnosed with HIV were excluded from participation. Can the authors explain why that was, given that one of the purposes of distributing HIVST kits is to offer sexual partners an opportunity to test?

Response 

We thank the reviewer for the comment. We fully agree with reviewer assertion that excluding those who tested HIV-positive on the same day may have missed an opportunity to get their partners tested. However, excluding patients newly diagnosed with HIV on the day of enrolment was motivated by the fact that those newly diagnosed patients might face significant psychological burden as they are presented with the life altering diagnosis. Asking these patients to consent to take part in the study would have exposed them to unnecessary emotional and psychological distress. Due to these reasons we decided to excluded them in the study.

We have added an explanation as shown in line 184-186 and this reads as follows “Given the emotional and psychological distress associated with a new HIV-positive diagnosis, those diagnosed with HIV on the day of visit were excluded.”

Comment 

In the “Data analysis” section (lines 182-184), the authors define “acceptability” as “the proportion of people invited to be PRs who accepted HIVST”. This is a bit confusing to me – is acceptability the number of people who agreed to take the tests and distribute them to others or is it the number of people who agreed to use the self-test?

Response 

We thank the reviewer for the comment and suggestion. We look at acceptability of peer distribution, rather than at acceptability of HIVST kit, hence, we define acceptability as the proportion of people who were approached and agreed to take HIVST kits for distribution. We have added the operational definition for acceptability as shown in line 172-174, “Acceptability was defined as the proportion of people invited to be PRs and agreed to take HIVST kits for use and distribute them to their sexual partner, family members or anyone in their social network." 

Comment 

Acceptability is typically defined as uptake – that is – are people willing to use the test. It has been pretty well established in the literature (including what is cited in this paper) that HIV self-testing is acceptable to most people in South Africa. I think the potential contribution of this paper is on the effectiveness (or not) of the distribution model for hard-to-reach populations in rural settings. That is, did it improve testing access or coverage in community and/or clinics settings? What evidence do the authors have to show that and how did it improve access and/or coverage specifically within the clinics (or were the kits taken outside of the clinic for broader distribution?) And beyond the community testing site? More on the how would be helpful.

Response 

We thank the reviewer for the comment and for the reflection around HIV self-testing in South Africa. We acknowledge the reviewer comment on the literature of HIV self-testing in South Africa recognising the acceptability of HIV self-testing in South Africa. However, there is limited evidence on the acceptability of unsupervised peer-distribution of HIV self-testing under programmatic conditions, particularly in HIV-high prevalence and low-income rural settings. Therefore, this paper addressed this research gap. Furthermore, this paper is of particular significance since it is a demonstration study assessing an acceptability of unsupervised peer-distribution of HIV self-tests or peer-led distribution as method to scale-up HIV testing in general population, under programmatic conditions rather than targeted interventions (FSWs, post-partum and pregnant women, fishing community, and MSM). 

Comment 

In Table 1 and in the description of the data (lines 195-201), it seems that the authors are describing acceptability as the number of people who were willing to take the HIVST kits and distribute them – is this correct? Also, the authors report on acceptability again in lines 207-208 showing a difference between community and primary health clinic sites. This is a bit confusing as most people think of this as how many people actually took the HIVST kit to test/use rather than the distributors. Is there a way to label this differently so it is clearer to the reader what the authors are describing and why that is important (e.g., the characteristics of the distributors?) or is this describing the characteristics of the people who actually tested with the HIVST kits?

Response 

We appreciate the reviewer’s comment. Under recruitment section, line 180-183 we have stated that “Individuals who expressed interest were given detailed information about the study and screened for eligibility. After informed consent, they were enrolled in the study as PRs and given between 2-5 HIVST kits, based on their choice, for distribution and/or self-testing.” We further stated in line 190-193 that “PRs were instructed to distribute HIVST kits to their sexual partner, family members or anyone in their social network who was above 18 years old. They could also test themselves if they were HIV-negative.” These two statement highlight that once an individual was enrolled in the study as primary recipients would have not only distributed HIVST kits but would have also tested themselves if they were HIV-negative. 

Furthermore, to avoid confusion we have rephrased and moved the following (previously line 207-208) “Acceptability of HIVST was thrice as high at community-based testing sites than at PHC (63.5% vs. 21.0%; p=0.001). The median age was similar in both community-based testing sites and PHCs (p=0.389).” to “Acceptability was thrice as high at community-based testing sites than at PHC (63.5% vs. 21.0%; p=0.001)” this is shown in line 250-252. 

Comment 

The authors report in lines 221-227 that of 34,715 adults that were tested for HIV at both primary care clinics and community-based testing sites during the study period, only 1,089 (3%) reported testing with an HIVST. This number is quite low and suggests potentially low coverage of distribution of the >31,000 HIVST kits or potential low uptake/use of the kits given the prevalence in the region and undiagnosed persons with HIV. The authors should consider why the number overall is low and discuss potential reasons for this in the Discussion section.

Response 

We thank the reviewer for this comment and concur that only 1,089 out 34,715 or 3.1% of individuals reported having used HIVST and this could mean low coverage distribution of the 31,341 in the study setting. We have acknowledged the low coverage distribution as one of many study limitations as shown in line 405-408 and this reads as follows “An inherent limitation of many studies of unassisted self-testing is the difficulty of verifying use and accuracy of the tests. In our study all information in the follow-up component was obtained through participants’ reports, and thus we were unable to verify use, or results.” 

Additionally, the small number of participants reporting HIVST use does not necessarily translate in higher proportion of undiagnosed considering our study setting is characterized by a high HIV testing coverage. A population based cross sectional survey done in 2018 reported 96.7% of the individuals surveyed having at least one HIV test done . 

Comment 

• In this same section, the authors report that the majority of the reported users of the HIVST kits were the distributors (primary recipients – 59%) who had mostly been tested within the past 6 months. This begs the question then of how this model is improving access to testing and coverage since most of these folks may likely be retesting.

Response 

We appreciate the reviewer’s comment. We fully agree with the reviewer’s comment. This study was not a surprising result considering the high HIV testing coverage in the study setting (please see previous response). We have extensively commented on this as shown in lines 344-355 and this reads as follows “During the study period, 1,089 individuals reported HIVST use; of these 58.8% tested themselves. Thus, not only did people agree to distribute HIVST, but they also used it as a method to test themselves in private. This can help, along with reaching more people, to also increase HIV testing frequency. Furthermore, a higher proportion of HIVST users had tested less six months before and a smaller proportion were first-time testers. This is not surprising considering the high HIV testing coverage in the study area [22]. Further research is needed on how HIVST will co-exist within HTS and on the role of HIVST in settings with high HIV testing coverage.”

Comment 

• The data in Figure 1 are a bit misleading. Specifically, in the column on the left it seems that the 36 confirmed positive persons whose records could not be found should not be removed from the denominator in the last box as these were confirmed positives but did not link or initiate ART. Similarly, the 25 already on ART seem to belong in the denominator as well since they are on ART. If the 25 are added back in, the number initiated or on ART would be 154/196 confirmed positives (79%) with 36 having no record of initiating ART and 6? This is a more accurate estimate of the number of PLHIV who initiated or were on treatment among the 196 HIVST PLHIV identified. The current estimate is an overestimate.

Response 

We appreciate the reviewer’s comment. We have added 36 confirmed HIV positive whose records could not be found in the denominator, this is shown Figure 1. However, 25 confirmed HIV positive patients who were already on ART before receiving HIVST could not be added in the denominator since they were not eligible to use HIVST. The revised text is shown in line 317-326 and this reads as follows “Of 196 (100%) confirmed HIV positive through HTS, clinical records of 36 (18.4%) participants could not be found after extensive search of electronic registers for the study sites (Figure 1). These were categorized as not linked to care. In addition, 25/160 (15.6%) were already on ART before receiving HIVST with median time in care of 53.2 months (IQR: 32.8-68.6), 129/171 (75.4%) initiated ART, whereas 2,362/2685 (88%) of HIV positive HIVST non-users-initiated ART.” 

We have commented in the discussion section about those known HIV-positive and using HIVST in line 374-381 and this reads as follows “Over 10% of those confirmed HIV positives had been diagnosed prior to study enrolment, despite clear instruction that the test should not be used by PLHIV on ART and warning of false positive HIVST if on ART [6, 29]. Repeat testing by those on ART has been reported elsewhere [28, 30] and suggests that PLHIV desire to re-confirm their HIV status for themselves, despite their reluctance to disclose their status. Programs should spend more time on patient education so that those on ART must no use HIVST. ”

Comment 

As shown above, the HIVST users actually had lower treatment initiation or on ART rates than those tested through other methods. How to address this issue and strengthen linkage to care for HIVSTers should be addressed in the Discussion section.

Response 

We appreciate the reviewer’s comment. We have written a paragraph relative to ART initiation among HIV user as shown in line 382-394 and this reads as follows “An important contribution of this study was an attempt to establish the proportion of individuals who returned for confirmatory testing and ART uptake following HIVST. In our study, confirmatory testing and ART uptake was lower among HIVST users compared to HIVST non-user, 75.4% and 88.0% respectively. The universal test and treat (UTT) policy might have contributed to this difference. UTT interventions are intensive testing programmes designed to find even the hardest-to-reach populations. Systematic review and meta-analysis of linkage to treatment or care following HIVST in the general population has found limited but encouraging evidence of successful linkage among HIVST users [31]. Further research is needed to assess cost-effective and sustainable interventions designed to strengthen linkage to treatment and care following HIVST, such as assisted referrals and home ART initiation. Additional data is needed to assess the effect on HIVST on linkage to prevention services among those who are HIV-negative and at risk under, particularly in HIV-high prevalence and low-income rural settings.”

Comment 

The Discussion section often repeats the results. Suggest that the authors briefly summarize the key findings without restating the data and offer information to help interpret the finding or discuss how it supports or contradicts the extant literature.

Response 

We thank the reviewer for the comment and have edited the discussion section as suggested. 

Comment 

Overall, this study has potential to propose a model for distribution of HIVST kits for use by hard-to-reach populations in rural settings in South Africa. However, there are current limitations in the way the data is presented and the lack of clarity of the findings that limit the interpretation of the data and its current contribution to the literature. The authors are encouraged to present the data in a way that would clearly show who is being reached with these test kits and if they are truly increasing access to testing in these populations (especially men and young people which appear to account for about a third of those using HIVST kits) or if they are, for example, primarily providing retesting options for those who are already testing regularly every 6 months.

Response 

We thank the reviewer for the comment. We would like to emphasise that the main objective of this study was to describe acceptability of unsupervised peer-distribution of HIVST as method to scale-up HIV testing in the general population, particularly under programmatic conditions. We believe that innovative delivery models of HIVST provide an opportunity to reach the hard-to-reach population. The study has shown that 26.9% of people approached to be peer-distributor or primary recipients accepted. However, the distribution model could not reach the hard-to-reach such men, young adults, and first-time testers to the extent as was anticipated. 

We have made several adjustments throughout the manuscript to address your comment. For instance, 422-430 acknowledge failure of this distribution model to not reach the hard-to-reach. However, the results show that taking HIVST kits outside of the conventional health facilities to community-based testing sites resulted in higher acceptability (please see line 399-402 and 422-430).

Comment 

The paper needs a copyediting review before it is re-submitted as there are a number of missing words, spaces, and punctuation.

Response 

We appreciate the reviewer’s comment. The paper has undergone major editing. 

Once again, we would like to thank the reviewers for the valuable comments and inputs that have helped to improve the quality of our proposal. 

We trust you will find the revised document and the responses in order.

Yours Sincerely 

Marcel Kanyinda Kitenge

---

## [Editor Report · Decision Letter 1]

11 Feb 2022

Acceptability of unsupervised peer-based distribution of HIV oral self-testing for the hard-to-reach in rural KwaZulu Natal, South Africa: Results from a demonstration study

PONE-D-21-19545R1

Dear Dr. Marcel Kanyinda Kitenge,

We’re pleased to inform you that your manuscript has been judged scientifically suitable for publication and will be formally accepted for publication once it meets all outstanding technical requirements.

Kind regards,

Limakatso Lebina, MBChB, Ph.D.

Academic Editor

PLOS ONE

Additional Editor Comments (optional):

You have managed to address all the comments from the reviewers and revised the manuscript accordingly.
---

## [Editor Report · Acceptance letter]

22 Mar 2022

PONE-D-21-19545R1 

Acceptability of unsupervised peer-based distribution of HIV oral self-testing for the hard-to-reach in rural KwaZulu Natal, South Africa: Results from a demonstration study 

Dear Dr. Kitenge:

I'm pleased to inform you that your manuscript has been deemed suitable for publication in PLOS ONE. Congratulations! Your manuscript is now with our production department. 

Kind regards, 

on behalf of

Dr. Limakatso Lebina 

Academic Editor

PLOS ONE